# Left ventricular diastolic dysfunction in non-severe chronic obstructive pulmonary disease – a step forward in cardiovascular comorbidome

**Zheina Cherneva**[1] *, **Dinko Valev**[2], **Vania Youroukova**[3], **Radostina Cherneva**[3]

**1** Medical Institute of the Ministry of Internal Affairs, Sofia, Bulgaria, **2** University First Multiple Clinic for Active Treatment, Sofia, Bulgaria, **3** University Hospital for Respiratory Diseases"St. Sophia", Sofia, Bulgaria

* jenicherneva@yahoo.com

**Data Availability Statement:** All relevant data are within the manuscript and its Supporting Information files.

## Abstract

Chronic obstructive pulmonary disease (COPD) augments the likelihood of having left ventricular diastolic dysfunction (LVDD)–precursor of heart failure with preserved ejection fraction (HFpEF). LVDD shares overlapping symptomatology (cough and dyspnea) with COPD. Stress induced LVDD is indicative of masked HFpEF. Our aim was to evaluate the predictive value of inflammatory, oxidative stress, cardio-pulmonary and echocardiographic parameters at rest for the diagnosis of stress LVDD in non-severe COPD patients, who complain of exertional dyspnea and are free of overt cardiovascular diseases. A total of 104 COPD patients (26 patients with mild and 78 with moderate COPD) underwent echocardiography before cardio-pulmonary exercise testing (CPET) and 1–2 minutes after peak exercise. Patients were divided into two groups based on peak average E/e': patients with stress induced left ventricular diastolic dysfunction (LVDD)—E/e' > 15 masked HFpEF and patients without LVDD—without masked HFpEF. CPET and echocardiographic parameters at rest were measured and their predictive value for stress E/e' was analysed. Markers for inflammation (resistin, prostaglandine E2) and oxidative stress (8-isoprostanes) were also determined. Stress induced LVDD occurred in 67/104 patients (64%). Those patients showed higher VE/VCO2 slope. None of the CPET parameters was an independent predictor for stress LVDD.Except for prostglandine E2, none of the inflammatory or oxidative stress markers correlated to stress E/e'. The best independent predictors for stress LVDD (masked HFpEF) were RAVI, right ventricular parasternal diameter and RV E/A >0.75. Their combination predicted stress LVDD with the accuracy of 91.2%. There is a high prevalence of masked HFpEF in non-severe COPD with exertional dyspnea, free of overt cardiovascular disease. RAVI, right ventricular parasternal diameter and RV E/A >0.75 were the only independent clinical predictors of masked HFpEF. 288.

**Funding:** The author(s) received no specific funding for this work.

**Competing interests:** The authors have declared that no competing interests exist.

## Introduction

Chronic obstructive pulmonary disease (COPD) patients frequently suffer from comorbidities [1]. COPD patients have 2–3 fold elevated predisposition of cardio-vascular (CV) events even when confounders are taken into account [2]. CV comorbidity in COPD is assumed as "cardio-pulmonary continuum" rather than being attributed to shared risk factors [3]. Cardio-respiratory interactions are not restricted to definite structural, haemodynamic, vascular or genetic parameters and both diseases are linked to systemic inflammation.

COPD augments the likelihood of having cardio-vascular diseases (CVD), the strongest association, being with heart failure [4]. The diagnosis of heart failure with preserved ejection fraction (HFpEF) in COPD is difficult. Its precursor—abnormal left ventricular relaxation, is termed left ventricular diastolic dysfunction (LVDD). It may be present, regardless of left ventricular ejection fraction (LVEF) or patient's symptoms [5,6].

The early detection of LVDD is an important part in the evaluation of COPD patients, as it can lead to heart failure and worse prognosis. The overlapping symptoms (dyspnea or chest pain) deter the timely diagnosis of the comorbidity (concomitant cardiac or pulmonary disease). Both COPD and heart failure exacerbations present similarly, making it difficult to distinguish one from another. Echocardiography is the key diagnostic modality for identifying diastolic dysfunction. The simultaneous performance of stress-echocardiography and cardio-pulmonary exercise testing may helpfully provide timely detection of early stages of LVDD in COPD patients with exertional dyspnoea. Their execution is, however, time consuming and demands special equipment.

With these objectives we set the following aims: 1) to detect the frequency of stress LVDD —masked heart failure with preserved ejection fraction (HFpEF) in non-severe COPD patients, free of overt cardiovascular pathology who complain of exertional dyspnea; 2) to establish which echocardiographic parameters at rest may be predictors for stress LVDD; 3) to establish which inflammatory (hsCRP, resistin, prostaglandine E2) and oxidative stress (8-iso-prostane) markers are predictors for stress LVDD.

## Materials and methods

### Patients and study protocol

It was a prospective study that was performed in 224 clinically stable outpatients, diagnosed with COPD at the University Hospital for Respiratory Diseases "St. Sophia", Sofia. Only 163 of them met the inclusion criteria: The inclusion criteria are: 1) non-severe COPD (post bronchodilatator FEV1/FVC<70%; FEV1/ > 50%); 2) preserved left ventricular systolic function LVEF>50%; 3) lack of overt cardiovascular disease; 4) exertional dyspnea. The flowchart of the study is presented in Fig 1.

All the subjects had exertional dyspnoea, but a total of 104 patients (64 men, 40 women; mean age of 62.9±7.5 years) were considered eligible, assuming the exclusion criteria. The recruitment period was between May 2017–April 2018, and was approved by the Committee of Ethics of Science of the Medical University, Sofia (protocol 5/12.03.2018). All the patients were preliminary acquainted with the aim of the study, its scientific value and the potential presentation of data at different forums. No minor participants were included. All of the participants delivered a written form of the protocol and objective of the study and signed their informed consent. We declare no conflict of interest regarding this study.

The following exclusion criteria were considered: 1) left ventricular ejection fraction (LVEF) < 50%; 2) left ventricular diastolic dysfunction at rest more than first grade; 3) presence of echocardiographic criteria of pulmonary hypertension (systolic pulmonary arterial

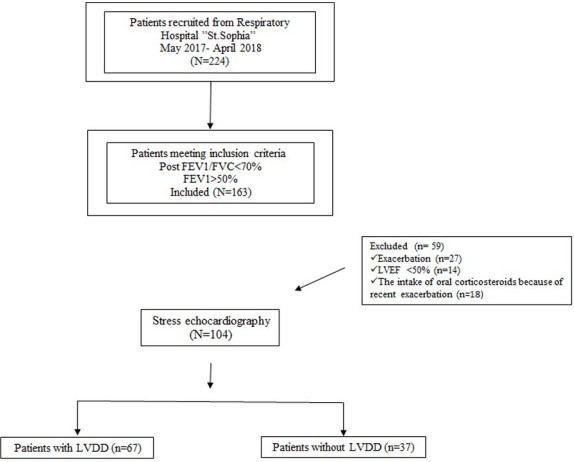

**Fig 1. Flowchart of the study protocol.**

pressure > 36 mmHg, maximum velocity of the tricuspid regurgitation jet > 2.8 m/s; 4) valvular heart disease–was excluded by the absence of structural and functional abnormalites at both rest and stress echocardiography; 5) documented cardiomyopathy; 6) severe uncontrolled hypertension (systolic blood pressure > 180 mmHg and diastolic blood pressure >90 mmHg); 7) atrial fibrillation or malignant ventricular arrhythmia; 8) coronary artery disease was excluded by ECG–none of the patients had angina symptoms at rest or during load; none of them showed ST-T changes during the load or in the recovery phase; 9) anaemia was excluded if Hb<100g/l; 10) diabetes mellitus; 11) cancer; 12) chronic kidney disease was excluded if eGFR <60ml/min; 13) recent chest or abdominal surgery; 14) recent exacerbation (during the last three months); 15) recent change (during the last three months) in inhalatory therapy.

Thirty-nine patients (32.5%) were excluded because of recent exacerbation; twenty-eight patients (23.3%) had a recent change in the concomitat inhalatory therapy; eight (7%) had anemia; twenty-four (20%) patienst had diabetes; five (4.2%) patients had chronic kidney failure.

## Procedures

### Pulmonary function testing

All the subjects undewent preliminary clinical examination which included chest X-ray, spirometry, electrocardiogram, echocardiography. Those eligible for the study performed spirometry and exercise stress test. They were performed on Vyntus, Cardio-pulmonary exercise testing (Carefusion, Germany) in accordance with ERS guidelines [7]. Only patients with mild/ moderate airway obstruction (FEV1 >50%) were selected.

### Dynamic hyperinflation (DH)

Body plethysmography (residual volume (RV), functional residual capacity (FRC), total lung capacity (TLC)) was performed on (Vyntus, body plethysmograph, CareFusion, Germany) using European and American Thoracic Society guidelines [7]. Changes in operational lung volumes were derived from measurements of dynamic inspiratory capacity (IC), assuming that total lung capacity (TLC) remained constant during exercise [8,9]. This has been found to be a reliable method of tracking acute changes in lung volumes [8–10]. IC was measured at the end of a steady-state resting baseline, at 2 min intervals during exercise, and at end exercise. End-expiratory lung volume (EELV) was calculated from IC maneuvers at rest, every 2

minutes during exercise and at peak exercise (Vyntus). In these maneuvers, after EELV was observed to be stable over 3–4 breaths, subjects were instructed to inspire maximally to TLC. For each measurement, EELV was calculated as resting TLC minus IC, using the plethysmographic TLC value. Dynamic IC (ICdyn) was defined as resting IC minus IC at peak exercise [11]. Dynamic hyperinflation (DH) was defined as a decrease in IC from rest of more than 150 mL or 4.5% pred at any time during exercise [11].

## Stress test protocol–cardio-pulmonary exercise testing (CPET)

All the patients underwent cardio-pulmonary exercise testing on a cycle ergometer. symptom limited incremental exercise stress test following the guidelines [12]. A continuous ramp protocol was applied. After two minutes of unloaded pedaling (rest phase- 0W), a three minute warm-up phase (20W) followed. The test phase included 20W/2min load increments. Patients were instructed to pedal with 60–65 rotations per minute. Patients' effort was considered to be maximal if two of the following criteria emerged: predicted maximal HR is achieved; predicted maximal work is achieved; 'VE/'VO2 >45, RER >1.10 as recommended by the ATS/ACCP [13]. The maximum HR (MHR) was calculated (MHR = 220—age). The target HR (THR) was set at 80% of MHR.

A breath-by-breath analysis was used for expiratory gases evaluation. 'VO2 (mL/kg/min), 'VCO2 (L/min), 'VE (L/min) and PetCO2 (mm Hg) were collected continuously at rest and throughout the exercise test. Peak values of oxygen consumption and carbon dioxide production were presented by the highest 30-second average value, obtained during the last stage of the exercise test. Peak respiratory exchange ratio was the highest 30second averaged value between'VO2 and 'VCO2 during the last stage of the test. Resting PetCO2 was the 2-minute averaged value in the seated position prior to exercise, while the peak value was expressed as the highest 30-second average value obtained during the last stage of the exercise test. Ten-second averaged 'VE and VCO2 data, from the initiation of exercise to peak, were used to calculate the 'VE/'VCO2 slope via least squares linear regression. It has been shown to produce clinically optimal information compared with derivations excluding data past the respiratory compensation point [14]. 'VE/'VCO2 slope was calculated as a linear regression function using 10-s averaged values and excluding the non-linear part of the relationship after the respiratory compensation point (where nonlinear rise in 'VE occurred relative to 'VCO2 in the presence of decrease of end-tidal pressure of CO2. As the study group consisted of COPD patients a dual approach for the measurement of the anaerobic threshold (AT) was applied. Both V-slope method and the ventilatory equivalents method for 'VO2 and 'VCO2 were used. The modified Borg scale was applied for peak dyspnea and leg discomfort.

## Echocardiography methods

Good quality echocardiographic images could be acquired in all of our patients with mild and moderate (non-severe) COPD. Echocardiography included the generally applied approaches of M-mode, two-dimensional and Doppler echocardiography. Routine structural and haemodynamic indices of both chambers were measured following the guidelines [6]. The systolic function of the left ventricle was defined by Simpson's modified rule. The diastolic function of both ventricles was evaluated by the E/A ratio at rest [6]. As a more precise approach for diastolic dysfunction detection, tissue Doppler analysis was used. We used e' value as the average of medial and the lateral measurements for the mitral annulus. The four recommended variables for identifying diastolic dysfunction at rest and their abnormal cut-off values are: annular e' velocity, septal e' < 7 cm/sec, lateral e' <10 cm/sec; average E/e' ratio > 14; LA volume index > 34 mL/m2; and peak TR velocity > 2.8 m/sec. LV diastolic dysfunction is present if more than half of the available parameters meet these cut-off values. Grade I diastolic dysfunction is considered if: E/A<1;

DT>200msec; average E/e'<8. Grade II is assumed if: 1> E/A <2; 160> DT <200msec; average 8>E/e'<15. Grade III is assumed if: E/A >2; DT<160msec; average E/e'>15.Stress echocardiography was performed 1-2minutes after peak exercise. It was considered positive when all of the following three conditions are met during exercise: average E/e' > 14 or septal E/e' ratio > 15, peak TR velocity > 2.8 m/sec and septal e' velocity < 7 cm/sec [6].

RV systolic function was assessed using tricuspid annular plane systolic excursion (TAPSE) and tissue Doppler S peak velocity. RV wall thickness (RVWT) was measured from the subcostal view at the tip of the anterior tricuspid leaflet in end-diastole. Pulmonary pressure was calculated directly by sampling the tricuspid insufficiency and indirectly by the acceleration time (AcT) on pulmonary flow. To calculate the systolic pulmonary arterial pressure (sPAP) we used the simplified Bernoulli equation (P = 4[TRmax]2) taking peak TR velocity + right atrial pressure (RAP). RAP is assumed by the size of inferior vena cava (IVC) at rest and its distensibility during inspiration. Right atrium volume index (RAVI) was measured at right ventricular end-systole by Simpson's modified rule. Stress induced RV diastolic dysfunction was considered if stress induced average RV E/e' ratio > 6. The average e' value was the average measurement of the medial and lateral side of the tricuspid annulus for three beats. All parameters were measured 1–2 minutes after peak exercise in patients lying on bed closely situated to the ergometer cycle (supine position). All parameters were measured at end-expiration and in triplicate during different heart cycles [15,16].

## Laboratory assays

Approximately 7 mL of venous blood was obtained from all cases. Blood samples were centrifuged immediately after collection and isolated plasma was stored in vials at –80˚C until assayed. Resistin was measured by commercial kits, following the procedure protocol. Resistin was determined by an ELISA kit (RayBio_ Human Resistin ELISA Kit Protocol (Cat#:ELH-Resistin-001) The intra- and interassay coefficients of variation in this assay kit ranged from 10 to 12%. Plasma resistin levels were measured in ng/ml.

## High Resolution Accurate Mass (HRAM) of 8-isoprostane and prostgalndine E2

Approximately 20 mL of urine was obtained from all cases The levels of 8-isoprostane and prostgalndine E2 in urine samples were determined by HRAM (high resolution accurate mass) mass spectrometry on LTQ Orbitrap® Discovery (ThermoScientific Co, USA) mass spectrometer, equipped with Surveyor® Plus HPLC system and IonMax® electrospray ionization module. The analyses were carried out by stable isotope dilution method in negative ionization mode using HESI II (heated electrospray ionization) source type. The concentration and purification of 8-isoprostane and prostgalndine E2 from urine samples was processed by affinity sorbent (Cayman Chemical, USA), following the producer's protocol with some modification. The urinary 8-isoprostane and prostgalndine E2 levels were standardized to the levels of urinary creatinine. Creatinine was measured applying the enzyme method—Creatinine plus version 2 Cobas Integra (Roche). Results are given in pg/mkmol/creatinine.

## Statistical analysis

Descriptive statistics was used for demographic and clinical data presentation. The Kolmogorov-Smirnov test was used to explore the normality of distribution. Continuous variables in each group of subjects were expressed as median and interquartile range when data was not normally distributed and with mean ±SD if normal distribution was observed. Categorical variables were presented as proportions. Data were compared between patients with and without

LVDD. An unpaired Student's t test was performed for normally distributed continuous variables. Mann-Whithney-U test was used in other cases. Categorical variables were compared by the χ2 test or the Fisher exact test. Correlation analysis was performed between cardio-pulmonary, echocardiographic, oxidaive stress and inflammatory markers and stress LVDD. Receiver operating characteristic (ROC) curves, a statistical technique used to determine parameter ability to discriminate between "gold standard normal and abnormal" were constructed. In our study ROC analysis was performed to test echocardiographic parameters at rest that may best accurately distinguish between stress LV E/e' >15 or < 15. The cut-off values with the best sensitivity and specificity were selected. Multivariate linaer regression analysis was also applied with those cardio-pulmonary, inflammatory and echocardiographic parameters (the echocardiographic parameters were evaluated as qualitative parameters, using their cut-off values). Predictive models were constructed. Age, sex, height, weight (BMI), FEV1, ICdyn, LV diastolic dysfunction at rest were specifically included as co-variates, as all of these have been previously reported as pathogenetic factors for LVDD.

In all cases a p value of less than 0.05 was considered significant as determined with SPSS® 13.0 Software (SPSS, Inc, Chicago, Ill) statistics.

## Results

### Demographic and clinical data

Subjects enrolled in the study were Caucasians at a mean age of 62.50±8.5 years and a body mass index of 27.26±6.92kg/m$^2$. They were divided into two groups—subjects with stress LVDD—64% (67/104) and those without—36% (37/104). There was no difference regarding the demographic, and respiratory parameters. The distirbution of COPD severity did not predominate in any of the subjects. The two groups, however, distinguished in some of their CPET parameters (Table 1).

### Cardio-pulmonary exercise testing parameters and stress LVDD

According to the objective ATS/ACCP criteria, exercise was considered maximal in all patients [13]. Patients differed significantly regarding the exercise cessation factors (Table 1). In patients with stress LVDD dyspnea was the predominant limiting factor—65 (97%). Leg fatigue was reported by 2(3%) of the patients with stress LVDD group (Table 1). The ventilatory and cardiovascular response parameters during exercise in the two groups are presented in Table 1 Most of the patients without stress LVDD 24 (65%) stopped exercise due to leg fatigue and only 13 (35%) reported of dyspnea. The two groups exhibited similar work load per body weight. The subjects without stress LVDD performed with lower VE/VCO2 slope. Despite of demonstrating higher (minute ventilation at peak load, higher oxygen pulse, higher peak 'VO2) these parameters were not statistically significant in comparison to the stress LVDD group.

### The prevalence of dynamic hyperinflation in patients with and without stress LVDD

None of the patients in the studied group demonstrated static hyperinflation. There was an even distribution of of hyperinflators/nonhyperinflators–among the patients with stress LVDD and those without (Table 1).

### LV parameters

Our patients were with normal LV dimensions and had preserved LV systolic function Table 2. The left atrial and ventricular dimensions were within normal limits and similar

**Table 1. Anthropometric, clinical, cardio-pulmonary parameters and biomarkers of the patients with and w/o stress LVDD.**

| | Patients w/o stress LVDD (37) | Patients with stress LVDD (67) | p-value |
|---|---|---|---|
| **Demographic data** | | | |
| Age, year | 60.00 ± 7.00 | 64.00 ± 7.00 | 0.143* |
| Male:Female gender, n | 21:16 | 44:23 | 0.298‡ |
| Current smokers, n (%) | 23(62%) | 39(58%) | 0.176‡ |
| Former smokers, n (%) | 4(11) | 17 (25) | 0.981‡ |
| Non-smokers, n (%) | 10(27) | 11 (17) | 0.375‡ |
| Packet years | 27.21 (23.87–31.76) | 33.79 (30.51–37.87) | 0.491† |
| Body mass index, kg/m2 | 27.00 (24.75–31.00) | 27.96 (22.75–30.75) | 0.207† |
| **Respiratory function** | | | |
| FVC, l/min | 2.06 (1.76–3.09) | 2.34 (1.77–3.09) | 0.213† |
| FEV 1, l/min | 1.31 (0.94–1.53) | 1.36 (1.14–1.75) | 0.408† |
| FEV1/FVC % | 60.5 (46.91–67.47) | 53.30 (45.76–66.55) | 0.764† |
| mMRC | 1.55 ± 0.49 | 1.70 ± 0.79 | 0.891† |
| **Acid-base balance** | | | |
| pO2, mmHg | 68.60(63.4–71.8) | 71.35 (64.7–74) | 0.298† |
| pCO$_2$, mmHg | 32.30 (30.1–35.37) | 37.65 (32.5–40) | 0.275† |
| Sat, % | 94.9 (94.4–95.25) | 95.00 (94.02–95.67) | 0.763† |
| **CPET parameters** | | | |
| Peak Load, W/kg | 1.14 (0.97–1.23) | 1,01 (0.91–1.22) | **0.529†** |
| Peak 'VE, l/min | 40 (34–52.5) | 38.50 (32–48) | 0.148† |
| Peak 'VO$_2$, ml/kg/min | 14.30(12.6–16.15) | 13.90 (12.67–15.7) | 0.794† |
| Peak RER | 1.06 (0.98–1.19) | 1.09 (1.00–1.28) | 0.808† |
| PeakO$_2$ pulse ml/kg/min | 9.80 (9.5–12.2) | 7.90 (6.15–9.32) | 0.751† |
| VE/VCO$_2$ slope | 34.08 (33.98–36.72) | 36.93 (34.19–38.74) | **0.032†** |
| **Exercise cessation factors** | | | |
| Dyspnea | 13(35%) | 65(97%) | 0.023‡ |
| Leg fatigue | 24(65%) | 2(3%) | 0.038 ‡ |
| **GOLD stages** | | | |
| GOLD I, n (%) | 9 (24%) | 17 (25) | 0.453‡ |
| GOLD II, n (%) | 28 (76%) | 50 (75%) | 0.814 ‡ |
| **Dynamic hyperinflation** | | | |
| ICdyn>150ml | 28 (76 6%) | 5176%) | 0.228‡ |
| ICdyn<150ml | 9(24%) | 16(24%) | 0.9971‡ |
| **Biomarkers** | | | |
| 8-isoprostane, mol/l/cre | 32.91±3.83 | 31.67±3.34 | 0.079* |
| hsCRP, mg/l | 3.4±0.8 | 3.6±0.1 | 0.063* |
| PG E2, mol/l/cre | 57.07±4.67 | 50.76 ±3.55 | 0.012* |
| Resistin, ng/ml | 22.51±2.61 | 19.68±3.56 | 0.847* |

*Unpaired t test;

†Mann-Whitney U test;

‡ chi square test;

§ Abbreviations: LVDD: Left ventricular diastolic dysfunction; GOLD–Global Initiative On Obstructive Lung Disease; O$_2$ pulse–oxygen pulse; 'VE–minute ventilation; RER–respiratory exchange ratio; 'VO$_2$ –oxygen consumption; FEV1 –Forced Expiratory Volume in 1s; FVC–Forced Ventilatory Capacity; mMRC- modified Medical Research Council; PG E2 –prostaglandin E2.

between groups. Only 30% of the patients had LV first grade diastolic dysfunction at rest (average E/e'<8); A total of sixty-seven percent (67%) of all the patients had LVDD during exercise

**Table 2. Echocardiographic parameters of the patients with and w/o stress LVDD.**

| | Patients w/o stress LVDD (37) | Patients with stress LVDD (67) | p-value |
|---|---|---|---|
| **LV structural parameters** | | | |
| LAVI, ml/m2 | 28.34(26.58–31.29) | 29.18(27.61–32.83) | 0.286* |
| TDD, mm | 50 (47.5–53) | 52 (48–55) | 0.506* |
| TSD,mm | 32 (28–35) | 34 (30–37) | 0.463* |
| TDV, ml | 120 (110–130) | 122.5(115–142) | 0.626* |
| TSV, ml | 39(37–43) | 42 (39–44) | 0.461* |
| LVEF, %, Simpson | 63.50(60–66) | 60.00(57–65) | 0.673* |
| Septum, mm | 12.00(11–13) | 12.00(11–13) | 0.897* |
| PW, mm | 12.00(11.75–12) | 12.00(11–13) | 0.981* |
| **LV functional parameters at rest** | | | |
| E/A ratio | 0.79(0.75–0.85) | 0.85 (0.76–1.20) | 0.420* |
| E/e' aver ratio | 6.66 (6.25–8.33) | 6.97 (5.76–8.15) | 0.736* |
| **LV functional parameters after exercise stress test** | | | |
| E/A ratio | 1.25(0.8–1.5) | 1.73 (1.55–2.00) | 0.042* |
| E/e' aver | 8.07 (6.7–9.6) | 17.33 (15.71–8.46) | 0.038* |
| **RV structural parameters** | | | |
| RAVI, ml/m2 | 17.57 (16.07–19.97) | 22.66 (21.31–24.13) | 0.037* |
| RVWT, mm | 5.00 (4.5–6.5) | 6.50 (6–7) | 0.046* |
| RV diameter parasternal, mm | 23 (21–25) | 28 (26–31) | 0.048* |
| RV diameter basal, mm | 35 (32–36) | 37(35.5–38) | 0.136* |
| RV diameter med, mm | 24 (22–26.75) | 26 (24.5–29) | 0.625* |
| **RV functional parameters at rest** | | | |
| E/A ratio | 0.83 (0.75–0.95) | 0.69 (0.62–0.75) | 0.761* |
| E/e' aver | 5.47 (4.56–5.69) | 4.16(3.33–5.00) | 0.764* |
| TAPSE,mm | 23.00 (22.00–26.00) | 22.00 (21.00–23.00) | 0.985* |
| TR jet velocity, m/s | 2.16 (1.98–2.31) | 2.34 (2.04–2.42) | 0.618* |
| AcT, msec | 170 (163.75–180) | 170(160–180) | 0.737* |
| sPAP, mmHg | 26.00 (25–28) | 28.00 (25–30) | 0.839* |

*Mann-Whitney U test;

† Abbreviations:LVDD: Left ventricular diastolic dysfunction; LAVI–left atrium volume index; RAVI–right atrium volume index; RVWT–right ventricular wall thickness; PW–posterior wall; TAPSE–tricuspid annular plane systolic excursion; AcT–acceleration time.

(E/e'>15). No significant difference in both structural and functional parameters of the LV at rest may be discerned between the patients with and without stress LVDD (Table 2).

## RV parameters

There was not a significant difference between the two groups regarding functional (systolic and diastolic) parameters of the RV at rest. Right atrium volume index (RAVI), RV parasternal diameter and right ventricular wall thickness (RVWT) showed significant difference between the groups with/without stress LVDD (Table 2).

## Echocardiographic parameters and stress LVDD

Some of the echocardiographic parameters (LV E/A ratio at rest, right atrium volume index, right ventricular wall thickness, right ventricular parasternal diameter, right ventricular E/A

ratio at rest) demonstrated statistically significant association with stress induced LVDD (E/ e'>15). To find the best cut-off values of these parameters ROC curves were constructed.

The only functional parameter of the LV with clinical importance is E/A ratio at rest.From the right heart structural parameters—RAVI, RVWT and the RV parasternal diameter are the echocardiographic indicators with good sensitivity and specificity for stress induced LVDD (Table 3). Figs 2–4 show the AUC of RAVI, RV parasternal diameter and RV E/A ratio at rest.

Bivariate correlation analysis was performed with the selected cut-off values. Data is presented in Table 4.

RAVI showed the highest odd ratio, followed by RV parasternal diameter, RV E/A ratio— and RVWT. In multivariable logistic regression analysis with a forward step approach and covariates age, BMI and forced expiratory volume in 1 sec—RAVI, the RV parasternal diameter and RV E/A remained the independent predictors for stress LVDD. The combination of these three echocardiographic parameters predicts stress LVDD with the accuracy of 91.2%. This association was independent of LV diastolic dysfunction at rest (LV E/A at rest; LV E/e' at rest), lung function (FEV1), ICdyn, age, sex, and BMI, taken as covariates.

## Markers for inflammation and oxidative stress

Markers of oxidative stress and inflammation are given in Table 1.Only prostaglandine E2 correlated to stress LV E/e', but was not an independent predictor for it (Table 4).

## Ventilatory and cardio-pulmonary exercise testing predictors for stress LVDD

An association between stress LVDD and the peak 'VO2, 'VO2 at AT, VE/VCO2 slope, O2pulse, HRR and ICdyn was observed (Table 4). Multivariable linear logistic regression analysis takes as covariates age, BMI, FEV1, RV, FRC, RV/TLC, IC/TLC, E/e' at rest, E/A at rest in a forward stepwise approach (Table 5). The multivariable linear logistic regression analysis demonstrated that none of the parameters was independently associated with stress E/e' >15 (Table 5).

## Discussion

The major findings of our study are: 1) thers is a high frequency—64% (67/104) of stress LVDD / masked heart failure with preserved ejection fraction in non-severe COPD patients with exertional dyspnea was established; 2) some of the cardio-pulmonary exercise testing parameters (peak 'VO2, 'VO2 at AT, VE/VCO2 slope, O2pulse and ICdyn) are associated with stress LV E/e' but none of them is an independent predictor for it; 3) markers of oxidative stress and inflammation are not independent predictors for stress LV E/e'; 4) the combination of the three echocardiographic parameters—RAVI, RV parasternal parameter and RV E/A

**Table 3. Receiver operating characteristic curve analysis using cut-off values of the echocardiographic parameters.**

|  | Area under the curve | 95% CI | Cut-off value | Sensitivity | Specificity |
|---|---|---|---|---|---|
| LV E/A ratio at rest | 0.62 | 0.51–0.73 | 0.86 | 56.06% | 77.78% |
| RV parasternal diameter, mm | 0.79 | 0.69–0.90 | 25.5 | 83.33% | 72.22% |
| RVWT, mm | 0.57 | 0.48–0.76 | 5.07 | 78.34% | 58.36% |
| RAVI, ml/m2 | 0.88 | 0.82–0.93 | 19.67 | 84.79% | 82.37% |
| RV E/A ratio at rest | 0.80 | 0.71–0.89 | 0.75 | 75.76% | 83.33% |

Abbreviations:LV–left ventricular; RV–right ventricular; RAVI–right atrium volume index; RVWT–right ventricular wall thickness.

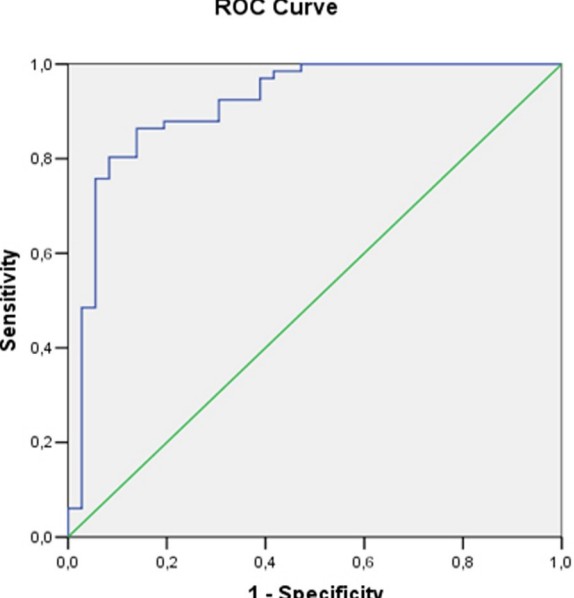

**Fig 2. Receiver operating curve analysis and area under the curve of RAVI.**

ratio may independently discern patients with stress LVDD from those without with the accuracy of 91.2%.The first systematic meta-analysis of diastolic dysfunction in COPD revealed that these patients are more likely to have LVDD [17]. The higher prevalence of LVDD in COPD population is a precondition to HFpEF. For the first time we applied combined exercise stress echocardiography in non-severe COPD patients with exertional dyspnea, free of overt

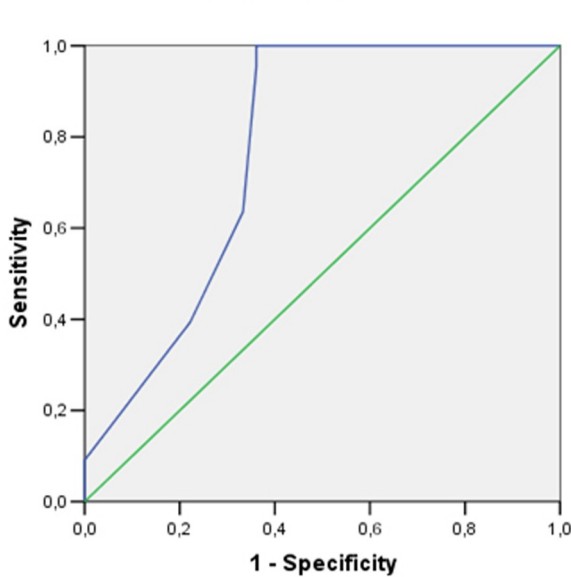

Diagonal segments are produced by ties.

**Fig 3. Receiver operating curve analysis and area under the curve of RV parasternal diameter.**

**Fig 4. Receiver operating curve analysis and area under the curve RV E/A ratio at rest.**

cardiovascular diseases. As most of the authors report on the incidence of diastolic dysfunction at rest we cannot compare our data to other studies of non-severe COPD patients [18,19]. The increase of E/e'>15 at peak exercise during the cardio-pulmonary testing is the cut-off point of left ventricular diastolic dysfunction (LVDD) and was considered as the marker of masked HFpEF in our patients. It was present in 67% o f them. Our results are much different from those in a non-COPD population. Nedeljkovic et al, report on 9.2% of masked HFpEF in hypertensive patients with exertional dyspnea and normal left ventricular function [20]. Kaiser et al, found that 9% of the patients with exertional dyspnea had E/A<0.75 [21].

The higher prevalence of masked heart failure with preserved ejection fraction in our COPD patients confirms the current notion that COPD is an independent predictor of vascular damage [22]. As most of the patients with LVDD are asymptomatic at rest, exercise reveals diastolic abnormalities even when they are not evident [23]. Stress echocardiography examines LV filling on exertion and detects the initial stages of diastolic dysfunction. Its performance is essential for the detection of diastolic dysfunction. This is of special clinical importance in COPD, where LVDD stays hidden under the umbrella of the COPD associated dyspnea. It may be an independent limiting factor of the physical activity and may influence COPD prognosis.

Our data shows that the echocardiographic parameters with the best predictive value for stress LVDD are RAVI, the parasternal diameter of the RV, and RV E/A ratio >0.75.

RAVI is a reproducible and easy to measure echocardiographic parameter that has gained interest during the last decade [24]. MRI and echocardiographic studies emphasize that right atrium geometry and RAVI are independent prognostic markers of heart failure with reduced ejection fraction (HFrEF) [25]. RAVI adds independent prognostic value to multifacet scores in which cardio-pulmonary parameters are components [26]. In HFrEF Sallach et al, and Darahim described modest correlation between RAVI and RV E/A ratio; no correlation however was found with E/e' ratio [24]. This confirms the poor correlation

**Table 4. Correlation analysis between stress LV E/e' ratio and the biomarkers, cardio-pulmonary parameters cut-off values of the echocardiographic idices.**

| Correlation analysis | p-value | correlation coefficient |
|---|---|---|
| **Echocardiographic parameters** | | |
| LV E/A ratio rest | 0.023 | 0.616 |
| RV parasternal diameter | 0.000 | 0.793 |
| RVWT | 0.000 | 0.219 |
| RAVI | 0.000 | 0.875 |
| RV E/A ratio rest | 0.000 | 0.417 |
| **CPET parameters** | | |
| Peak Load | 0.730 | 0.957 |
| Peak VE | 0.287 | 0.613 |
| V'O2 | 0.048 | 0.574 |
| AT, V'O2 | 0.021 | 0.216 |
| RER | 0.943 | 0.452 |
| VE/VCO2 slope | 0.026 | 0.612 |
| HR at rest | 0.737 | 0.247 |
| Peak HR | 0.382 | 0.409 |
| CRI | 0.061 | 0.752 |
| O2 pulse | 0.032 | 0.481 |
| HRR at 1 min | 0.041 | 0.763 |
| BR, % | 0.983 | 0.213 |
| ICdyn | 0.037 | 0.043 |
| **Biomarkers** | | |
| PG E2 | 0.041 | 0.038 |

Abbreviations: LV–left ventricular; RV–right ventricular; RAVI–right atrium volume index; RVWT–right ventricular wall thickness; PG E2 –prostglandine E2.

of right atrial volume to RV filling pressures [27]. Sallach et al, however report that RAVI is significantly associated to LV diastolic dysfunction. It is plausible that LVDD exacerbates pulmonary congestion and increases additionally the pulmo-capillary wedge pressures in patients with HFrEF. Both pulmo-capillary and pulmonary venous pressure elevation are transmitted retrogradely and manifest as RV overload and RA enlargement. The pathophysiology of right atrium remodeling that we present in our patients may be very similar to the mentioned above.

In COPD patients with HFpEF, transthoracic pressure gradients may additionally ccelerate right atrium/chamber remodeling and they may become apparent even at more early stages of LVDD than in the general population. This is confirmed by the fact that the COPD patients with stress LVDD have significantly changed RA geometry in comparison to those without stress LVDD. Having in mind that both LVDD and pulmonary hypertension are

**Table 5. Multivariate regression analysis between stress LV E/e' ratio and the cut-off values of the echocardiographic idices.**

| Multivariable logistic regression analysis | p-value | OR | 95% CI |
|---|---|---|---|
| RV parasternal diameter | 0.001 | 19.567 | 3.131–22.290 |
| RAVI | 0.000 | 24.061 | 4.485–29.100 |
| RV E/A ratio | 0.007 | 10.853 | 1.913–21.564 |

associated with increased number of exacerbations, accelerated decline of ventilatory function, and higher mortality, their timely detection is of clinical importance [28]. Whether left-sided dysfunction precedes or follows right-sided dysfunction in HFrEF is however elusive. The data regarding RAVI in COPD patients is described under the conditions of pulmonary hypertension and chronic respiratory failure. The pathophysiology of elevated RAVI in COPD patients is unresolved. The same is the issue regarding RAVI in HF. Having in mind the prognostic role and easy measurement of RAVI, catheterization studies are demanded to determine its precise pahophysiological role in impaired LV cardiac function and hemodynamics.

Though speculative systemic inflammation, oxidative stress, pulmonary hypertension, chronic hypoxemia, chronic hypercapnia, hyperinflation, and right-to-left ventricular interaction may all contribute to it. Systemic inflammation is a known contributor to the development of HFpEF. COPD itself causes elevation of IL-6, TNF-α, hsCRP. These proinflammatory cytokines increase E-selectin, VCAM, endothelial reactive oxygen species and attenuate nitric oxide availability in the coronary microvasculature [29]. Despite of analyzing the role of some biomarkers that are already associated with LVDD in the general population, none of these proved to be an independent predictor for it.

Both resistin and hsCRP are inflammatory markers that have been associated with vascular damage and increased cardiovascular morbidity [30,31]. In the general population resistin is being associated with LVDD and all the clinical conditions (diabetes, obesity, hypertension), predisposing to it. Despite this in our study its plasma levels were similar among COPD patients with/without stress LVDD. The only inflammatory marker that significantly differed between both groups was prostaglandin E2. It has been described as beneficial in cardiac remodeling after ischaemic injury [32,33]. We also describe data, supporting this notion. Urine levels of prostaglandin E2 are higher in the group without stress LVDD. They, correlated to stress LV E/e', but are not independent predictors for it. In addition to systemic inflammation, oxidative stress in COPD may also disturb calcium transport and myocardial relaxation [34]. The endothelial damage, caused by oxidative stress, affects both coronary, systemic and pulmonary vessels and exerts multifaceted mechanisms, that contribute to right (RVDD) and left ventricular diastolic dysfunction (LVDD)·[35]. Though we applied a well-validated method and marker for oxidative stress–urine 8-isoprostanes, we did not detect substantial difference in its concentrations between COPD subjects with/without LVDD. Neither a correlation between urine 8-isoprostanes and stress LV E/e' was found.

In conclusion, we report a high prevalence of masked HFpEF in non-severe COPD patients with exertional dyspnea, even if they are free of overt cardiovascular diseases. The combination of RAVI, right ventricular parasternal diameter, RV E/A>0.75 may predict masked HFpEF in these patients with 91% accuracy.

## Study limitations

The small sample size of the study is an apparent limitation, but it appears to be the first study to analyze the relationship between stress LVDD, CPET, echocardiographic parameters and biomarkers in non-severe COPD patients with exertional dyspnea; 2) the study is performed in a very selected group of COPD patients that are free of overt CV diseases e.g. (they do not show the common risk factors, associated with LVDD); 3) there is lack of standardisation of the way to perform stress echo after veloergometry (the standard position, time period during or after the load; the way to perform it–passively or during unloaded pedaling; lying on a bed or upright on the ergometer); 4) the results should be replicated in another cohort of non-severe COPD patients with exertional dyspnea in order to be validated.

## Supporting information

**S1 File.**
(XLSX)

## Acknowledgments

We give our acknowledgements to professor Vukov, who performed the statistical analysis.

## Author Contributions

**Conceptualization:** Zheina Cherneva, Vania Youroukova.

**Data curation:** Zheina Cherneva, Dinko Valev, Radostina Cherneva.

**Formal analysis:** Zheina Cherneva, Radostina Cherneva.

**Funding acquisition:** Dinko Valev.

**Investigation:** Zheina Cherneva, Vania Youroukova, Radostina Cherneva.

**Methodology:** Zheina Cherneva, Vania Youroukova, Radostina Cherneva.

**Project administration:** Vania Youroukova.

**Resources:** Dinko Valev, Vania Youroukova.

**Software:** Radostina Cherneva.

**Supervision:** Vania Youroukova.

**Validation:** Vania Youroukova, Radostina Cherneva.

**Visualization:** Vania Youroukova, Radostina Cherneva.

**Writing – original draft:** Zheina Cherneva, Radostina Cherneva.

**Writing – review & editing:** Vania Youroukova.

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
