## [Decision Letter · Decision Letter 0]

17 Aug 2020

PONE-D-20-21967

Left ventricular diastolic dysfunction in non-severe chronic obstructive pulmonary disease  – a step forward in cardiovascular comorbidome

PLOS ONE

Dear Dr. Cherneva,

Thank you for submitting your manuscript to PLOS ONE. After careful consideration, we feel that it has merit but does not fully meet PLOS ONE’s publication criteria as it currently stands. Therefore, we invite you to submit a revised version of the manuscript that addresses the points raised during the review process.

Please note that one reviewer asks for significant changes. I would like to highlight that you need to address them appropriately before the paper may be acceptable for publication.

We look forward to receiving your revised manuscript.

Kind regards,

Hans-Peter Brunner-La Rocca, M.D.

Academic Editor

PLOS ONE

Journal Requirements:

3. Your ethics statement must appear in the Methods section of your manuscript. If your ethics statement is written in any section besides the Methods, please move it to the Methods section and delete it from any other section. Please also ensure that your ethics statement is included in your manuscript, as the ethics section of your online submission will not be published alongside your manuscript.

Reviewers' comments:

Reviewer's Responses to Questions

**Comments to the Author**

1. Is the manuscript technically sound, and do the data support the conclusions?

Reviewer #1: Partly

Reviewer #2: Partly

2. Has the statistical analysis been performed appropriately and rigorously? 

Reviewer #1: Yes

Reviewer #2: Yes

3. Have the authors made all data underlying the findings in their manuscript fully available?

Reviewer #1: No

Reviewer #2: Yes

4. Is the manuscript presented in an intelligible fashion and written in standard English?

Reviewer #1: No

Reviewer #2: Yes

5. Review Comments to the Author

Reviewer #1: In the present paper the authors assessed the prevalence of stress-induced left ventricular diastolic dysfunction (LVDD) in 104 non-severe COPD patients, and looked at the value of biomarkers, resting echocardiography and CPET parameters to predict stress-induced LVDD. The authors have previously reported on this cohort. LVDD was defined as exercise E/e’ >15 (echo directly after CPET), and this was found in 67/104 (64%) patients. These patients had lower work rate, higher VE/VCO2 slope, higher right atrial volume index, right ventricular parasternal diameter, and RV E/A, the latter three being independent predictors. There was no association between biomarkers with LVDD except for PGE2 (slightly higher in those with LVDD, other measured markers: resistin, hsCRP, 8-isoprostane). The authors concluded that there is a high prevalence of LVDD in non-severe COPD that higher right atrial volume index, right ventricular parasternal diameter, and RV E/A were the only independent predictors.

General comment

The paper deals with an interesting topic but I see several problems. First, the terms LVDD and HFpEF are sometimes used synonymously, and sometimes not. It is not clear to me how the authors want to differentiate the two terms. Second, the authors have previously reported on this cohort, and there is a lot of overlapping data when comparing to previous articles (Cherneva R et al. Croat Med 2019, Cherneva R et al. Turk Kardiyol Dern Ars). It will be up to the editors to decide whether this overlap is prohibitive for another publication or not. Third, exercise LVEDD was defined as E/e’ >15 which is fine (more or less the criterion for a positive diastolic stress according to the 2019 ESC HFpEF definition paper) but it was very surprising to see that this did not correlated with resting E/e’ and LAVI. Thus, the results are quite surprising. Fourth, the authors state that this was a retrospective study. A protocol is used in the present study is typically not used in routine practice, and therefore I have doubts whether this was really a retrospective study. Measurement of E/e’ during/after exercise is very challenging (Obokata et al. Circulation 2017), and standardization is very important. Fifth, the paper is very hard to read because of its length, in part complex language, and redundant presentation of data in text and tables.

Specific comments

Abstract: the severity of COPD of the study patients should be specified. The terms LVDD and HFpEF should be separated, and only one should be used.

In line 10, the word “without” is missing: “…and patients LVDD (without HFpEF)..”

Patients and study protocol: the authors state that this was a retrospective study. What was the reason to establish such a protocol in clinical practice? The authors also report the recruitment period and ethical approval. I understand that this was a prospective study, correct?

Exclusion criteria: how was coronary artery disease excluded, how was valvular heart disease defined, how was chronic kidney disease defined, how was anemia defined?

CPET: the authors should clearly state that this a cycle ergometer test. How was predicted maximal heart rate determined? Achievement of predicted maximal work is not a criterion for maximal effort!

Echo: was stress echo performed when patients were sitting on the ergometer after completion of CPET, or were they transferred on a bed? Calculation of mPAP is not necessary for this study in my view.

Statistical analysis: the tests are appropriately described. Therefore it is not required to report the test applied for each parameter in the tables.

Results: this section should be shortened markedly. Data which are presented in Tables can be reported only qualitatively in the text.

Results/Table 1: there was a small difference in maximal work rate but not peak VO2 in patients with and without LVDD. The latter is in contrast to the statement in the abstract. The authors should express work rate indexed to body weight (peak VO2 is also indexed).

Results/Table 2: the difference in LAVI was not significant. The text is misleading. All abbreviations must be explained.

Table 3: there was no difference in LV E/A betwenn groups. So why is an AUC for E/A to predict LVDD constructed?

Table 4. Is this a logistic regression to find predictors of exercise LVDD? This should be clearly stated. Only parameters with significant p values in Tables 1 and 2 should be included.

Discussion: is too long, must be condensed.

References: too many, must be shortened to approximately 30. Refs 18 and 28 are the same ones.

Reviewer #2: Summary

Cherneva et al evaluated the predictive value of echo, CPET and some biomarkers for stress LVDD in non-severe COPD patients. A total of 104 COPD patient were evaluated. Stress LVDD was diagnosed as E/e’> 15 during exercise. In patients with increased E/e’ during exercise (1) the RA volume index was increased, (2) RV diameter parasternal, but not RV diameter basal, was enlarged and (3) RV wall thickness increased. RAVI, RV E/A and RV diameter basal were independent predictors for stress LVDD.

Comments:

Abstract/Introduction

• In abstract and in paper the terms stress LVDD and HFpEF are used interchangeable. This is incorrect. First exercise E/e’ has a limited sensitivity to diagnose HFpEF. Secondly, in the cohort studied LVDD at rest is excluded, as are patients with pulmonary hypertension at rest and AF patients. By excluding these patients, likely a large proportion of HFpEF patients are excluded. Consequently, these analyses have been performed in a very selected cohort of COPD patients.

Methods

• Since the list of exclusion criteria is extensive it is informative to state how many patients were excluded, and for which reason.

• Stress LVDD is in abstract described as E/e’>15, but in methods sections this diagnosis was considered when all of the following 3 conditions were present: average E/e’> 14 or septal E/e’ >15 AND peak TR >2.8, AND septal e’ velocity <7cm/sec. Which of these criteria did the authors use?

Results

• Table 1 contains incorrect data: GOLD stages: of the 67 patients with stress LVDD 72 were diagnosed as gold II. Similar mistake was probably made for dynamic hyperinflation.

Please re-analyze, and if differences are not statistically different (such as dynamic hyperinflation), then there is no ‘predominant prevalence’ in one of the groups.

• Authors conclude patients without stress LVDD had a better exercise tolerance, however this was only assessed in univariate assessment, it would be interesting to correct for age and COPD severity.

• If COPD severity is different between the groups (after providing the corrrect data, but based on the current data I think it might be), I would recommend including this in the multivariate analyses.

Discussion

• Minor: high frequency of LVDD = high frequency of stress LVDD

Additional comments

• An external validation to assess predictors of stress LVDD would strengthen the paper. I am especially sceptic about RV diameter as an independent predictor (as this is only when measured parasternal, but not the basal diameter)

6. PLOS authors have the option to publish the peer review history of their article (what does this mean?). If published, this will include your full peer review and any attached files.

Reviewer #1: No

Reviewer #2: No

---

## [Author Response · Author response to Decision Letter 0]

14 Sep 2020

E-D-20-21967

Left ventricular diastolic dysfunction in non-severe chronic obstructive pulmonary disease – a step forward in cardiovascular comorbidome

PLOS ONE

Dear Dr. Cherneva,

Thank you for submitting your manuscript to PLOS ONE. After careful consideration, we feel that it has merit but does not fully meet PLOS ONE’s publication criteria as it currently stands. Therefore, we invite you to submit a revised version of the manuscript that addresses the points raised during the review process.

Please note that one reviewer asks for significant changes. I would like to highlight that you need to address them appropriately before the paper may be acceptable for publication.

We look forward to receiving your revised manuscript.

Kind regards,

Hans-Peter Brunner-La Rocca, M.D.

Academic Editor

PLOS ONE

Journal Requirements:

3. Your ethics statement must appear in the Methods section of your manuscript. If your ethics statement is written in any section besides the Methods, please move it to the Methods section and delete it from any other section. Please also ensure that your ethics statement is included in your manuscript, as the ethics section of your online submission will not be published alongside your manuscript.

Reviewers' comments:

Reviewer's Responses to Questions

Comments to the Author

1. Is the manuscript technically sound, and do the data support the conclusions?

Reviewer #1: Partly

Reviewer #2: Partly

2. Has the statistical analysis been performed appropriately and rigorously?

Reviewer #1: Yes

Reviewer #2: Yes

3. Have the authors made all data underlying the findings in their manuscript fully available?

Reviewer #1: No

Reviewer #2: Yes

4. Is the manuscript presented in an intelligible fashion and written in standard English?

Reviewer #1: No

Reviewer #2: Yes

5. Review Comments to the Author

Reviewer #1: In the present paper the authors assessed the prevalence of stress-induced left ventricular diastolic dysfunction (LVDD) in 104 non-severe COPD patients, and looked at the value of biomarkers, resting echocardiography and CPET parameters to predict stress-induced LVDD. The authors have previously reported on this cohort. LVDD was defined as exercise E/e’ >15 (echo directly after CPET), and this was found in 67/104 (64%) patients. These patients had lower work rate, higher VE/VCO2 slope, higher right atrial volume index, right ventricular parasternal diameter, and RV E/A, the latter three being independent predictors. There was no association between biomarkers with LVDD except for PGE2 (slightly higher in those with LVDD, other measured markers: resistin, hsCRP, 8-isoprostane). The authors concluded that there is a high prevalence of LVDD in non-severe COPD that higher right atrial volume index, right ventricular parasternal diameter, and RV E/A were the only independent predictors.

General comment

The paper deals with an interesting topic but I see several problems. 

First, the terms LVDD and HFpEF are sometimes used synonymously, and sometimes not. It is not clear to me how the authors want to differentiate the two terms. 

Dear reviewer,thank you for the comments.We have made a mistake which is technical, but misleading. We mean that stress LVDD is indicative of masked HFpEF. We have gone and corrected this throughout the manuscript, because this is essential for the reader.

Second, the authors have previously reported on this cohort, and there is a lot of overlapping data when comparing to previous articles (Cherneva R et al. Croat Med 2019, Cherneva R et al. Turk Kardiyol Dern Ars). It will be up to the editors to decide whether this overlap is prohibitive for another publication or not. 

Dear reviewer you are completely right in your analysis. However, I hope that both the editors and you would appreciate the manuscript. Yes, to some extent there is an overlap of info.That is why the title is a step forward.We have addedd the biomarker analysis and have gone thoroughly through the whole pathogenesis of stress LVDD in non-severe COPD.

Third, exercise LVEDD was defined as E/e’ >15 which is fine (more or less the criterion for a positive diastolic stress according to the 2019 ESC HFpEF definition paper) but it was very surprising to see that this did not correlated with resting E/e’ and LAVI. Thus, the results are quite surprising.

Dear reviewer, thank you for the remark. We are speaking about exercise LVDD which according to the guidelines is considered positive if all of the following three conditions are met during exercise: average E/e’ > 14 or septal E/e’ ratio > 15, peak TR velocity > 2.8 m/sec and septal e’ velocity < 7 cm/sec.”

.‘’Recommendations for the Evaluation of Left Ventricular Diastolic Function by Echocardiography: An Update from the American Society of Echocardiography and the European Association of Cardiovascular Imaging. J Am Soc Echocardiogr 2016;29:277-314’’. So all the parameters that define stress LVDD are measured during exercise.

LVDD at rest is however defined as:

„The four recommended variables for identifying diastolic dysfunction at rest and their abnormal cut-off values are: annular e’ velocity, septal e’ < 7 cm/sec, lateral e’ <10 cm/sec; average E/e’ ratio > 14; LA volume index > 34 mL/m2; and peak TR velocity > 2.8 m/sec. LV diastolic dysfunction is present if more than half of the available parameters meet these cut-off values.‘’Recommendations for the Evaluation of Left Ventricular Diastolic Function by Echocardiography: An Update from the American Society of Echocardiography and the European Association of Cardiovascular Imaging. J Am Soc Echocardiogr 2016;29:277-314’’. So LVDD at rest is related to the parameters you are commenting on. 

From the citations above, I think our results are not surprising it is very common in diseases pathogenesis that the functional abnormalities precede the structural, this becomes especially evident when the organ is under extreme conditions.

Fourth, the authors state that this was a retrospective study. A protocol is used in the present study is typically not used in routine practice, and therefore I have doubts whether this was really a retrospective study. 

Dear reviewer,thank you for the comments. It is a stylistic mistake. It is a prospective cross-sectional study as you have perceived from the whole design.

Measurement of E/e’ during/after exercise is very challenging (Obokata et al. Circulation 2017), and standardization is very important.

Dear reviewer, thank you for the comments.You are right the results should be replicated in a separate cohort in order to be validated. We have commented on this in the limitation section, that is added to the manuscript as an additional section.

 Fifth, the paper is very hard to read because of its length, in part complex language, and redundant presentation of data in text and tables.

Dear reviewer, thank you for the comments, as you can see from the revised version the results section and the discussion have been extensively rewritten taking your recommendations in assumption.

Specific comments

1.Abstract: the severity of COPD of the study patients should be specified. The terms LVDD and HFpEF should be separated, and only one should be used.

In line 10, the word “without” is missing: “…and patients LVDD (without HFpEF)..”

Dear reviewer, thank you for the comments.We have added the severity distribution of COPD in the abstract.The missing word is also added. Regarding the issue LVDD and HFpEF we have already commented on that. 

2.Patients and study protocol: the authors state that this was a retrospective study. What was the reason to establish such a protocol in clinical practice? The authors also report the recruitment period and ethical approval. I understand that this was a prospective study, correct?

Dear reviewer, thank you for the comments. It is a stylistic mistake. It is a prospective cross-sectional study as you have perceived from the whole design.

3.Exclusion criteria: how was coronary artery disease excluded, how was valvular heart disease defined, how was chronic kidney disease defined, how was anemia defined?

Dear reviewer, thank you for the comments. The exclusion criteria have been precisely defined in the text. Regarding the coronary artery disease all patients that are included in the study underwent stress velo-ergometry and are without induced angina and ischaemic ECG changes. None of them had typical angina symptoms; none of them reported of angina symptoms during cardio-pulmonary exercise testing and none had ischaemic ST- T changes during the load or in the recovey phase. So none of them had clinical indications for coronary angiography. Some of these patients may possibly have non-obstructive coronary artery disease, but stress echocardiography or ECG during load are not the precise diagnostic approaches for its early detection.

CPET: the authors should clearly state that this a cycle ergometer test. How was predicted maximal heart rate determined? Achievement of predicted maximal work is not a criterion for maximal effort!` 

Dear reviewer, thank you for the comments.We have addedd in the text all the issues you pointed out – the cycle ergometer device and the formula for max.heart rate. Definitely the achievement of maximal work is not a predictor for maximal effort. We have pointed in the text the criteria for max effort.

‘’American Thoracic Society; American College of Chest Physicians. ATS/ACCP Statement on cardiopulmonary exercise testing. Am J Respir Crit Care Med.2003;167(2):211–277.’’

Echo: was stress echo performed when patients were sitting on the ergometer after completion of CPET, or were they transferred on a bed? Calculation of mPAP is not necessary for this study in my view.

Dear reviewer, thank you for the comments.We have added in the text the exact procedure of echo measurement. The patients were transferred to a closely situated bed and echo performed in lying position.We have deleted the description of mPAP from the manuscript.

Statistical analysis: the tests are appropriately described. Therefore it is not required to report the test applied for each parameter in the tables.

Dear reviewer, thank you for the comments,but the general reader may not have so sophisticated knoweledge in statistics, so we have presented data in this way to facilitate the reader.

Results: this section should be shortened markedly. Data which are presented in Tables can be reported only qualitatively in the text.

Dear reviewer,thank you for the comments, as you can see from the revised version the results section and the discussion have been extensively rewritten taking your recommendations in assumption.

Results/Table 1: there was a small difference in maximal work rate but not peak VO2 in patients with and without LVDD. The latter is in contrast to the statement in the abstract. The authors should express work rate indexed to body weight (peak VO2 is also indexed). 

Dear reviewer,thank you for the comments. I have addressed the issue in the abstarct and in the text as well.

Results/Table 2: the difference in LAVI was not significant. The text is misleading. All abbreviations must be explained.

Dear reviewer,thank you for the comments. I have corrected the issue in the text.

Table 3: there was no difference in LV E/A betwenn groups. So why is an AUC for E/A to predict LVDD constructed?

Dear reviewer, thank you for the comments.You are right there is no difference in LV E/A between groups. This is the reason for the poor performance of LV E/A AUC.We have made it to precisely compare the diagnostic value of the echo parameters and to point out that surprisingly LV E/A is worse than RV E/A at rest and other RV parameters when speaking of stress LVDD. This implicated that the haemodynamic consequences of stress LVDD reflect in RV structural and functional abnormalities first.

Table 4. Is this a logistic regression to find predictors of exercise LVDD? This should be clearly stated. Only parameters with significant p values in Tables 1 and 2 should be included.table.4.

Correlaton analysis between stress E/e’’and the other parameters was done. Multivaraible logistic models were built taking as covariates, parameters that are known to be responsible for LVDD pathogenesis.

Discussion: is too long, must be condensed.

Dear reviewer,thank you for the comments, as you can see from the revised version the discussion has been extensively rewritten taking your recommendations in assumption.

References: too many, must be shortened to approximately 30. Refs 18 and 28 are the same ones.

Dear reviewer,thank you for the comments. The refernces have been shortened to 35.

Reviewer #2: Summary

Cherneva et al evaluated the predictive value of echo, CPET and some biomarkers for stress LVDD in non-severe COPD patients. A total of 104 COPD patient were evaluated. Stress LVDD was diagnosed as E/e’> 15 during exercise. In patients with increased E/e’ during exercise (1) the RA volume index was increased, (2) RV diameter parasternal, but not RV diameter basal, was enlarged and (3) RV wall thickness increased. RAVI, RV E/A and RV diameter basal were independent predictors for stress LVDD.

Comments:

Abstract/Introduction

• In abstract and in paper the terms stress LVDD and HFpEF are used interchangeable. This is incorrect. First exercise E/e’ has a limited sensitivity to diagnose HFpEF. Secondly, in the cohort studied LVDD at rest is excluded, as are patients with pulmonary hypertension at rest and AF patients. By excluding these patients, likely a large proportion of HFpEF patients are excluded. Consequently, these analyses have been performed in a very selected cohort of COPD patients.

Dear reviewer,thank you for the comments. Stress LVDD is not interchangeable with HFpEF. We meant MASKED HFpEF as it is written in the aim of the study in the introduction section. Stress E/e’ is met in only a part of the patients with HFpEF. You are right that we have performed the study in a very selected group. The point of the manuscript is to show that stress LVDD in COPD may exist in patients that are free from the common risk factors mentioned above and the clinician (pulmonologist,cardiologist ) should suspect LVDD even in such COPD subjects. We have added your recommendations in a limitation section at the end of the manuscript. 

Methods

• Since the list of exclusion criteria is extensive it is informative to state how many patients were excluded, and for which reason. 

Dear reviewer,thank you for the comments. We have also added in the material and methods the number and the reason for the exclusion of participants

• Stress LVDD is in abstract described as E/e’>15, but in methods sections this diagnosis was considered when all of the following 3 conditions were present: average E/e’> 14 or septal E/e’ >15 AND peak TR >2.8, AND septal e’ velocity <7cm/sec. Which of these criteria did the authors use?

Dear reviewer, thank you for the comments. We have assumed that there is stress LVDD if all of the above mentioned criteria are met. We have used in the abstract only the average E/e’ ratio because it is the most important of them. If you suggest we may mention the three parameters.

Results

• Table 1 contains incorrect data: GOLD stages: of the 67 patients with stress LVDD 72 were diagnosed as gold II. Similar mistake was probably made for dynamic hyperinflation.

Dear reviewer, thank you for the comments. We have made a mechanistic mistake. The number of COPD grade I in stress LVDD is 17, not 7.We have corrected it. We have given percentage numbers as from the total number ot patients with and without stress LVDD, not as from the total number of all participants. The re-estimated percentages show that there is no predominance of the COPD grades in any of the two groups.

 Regarding the hyperinflators, we have also corrected the data and presented numbers as percentages within each group. (stress LVDD/w/t stress LVDD).

Please re-analyze, and if differences are not statistically different (such as dynamic hyperinflation), then t there is no ‘predominant prevalence’ in one of the groups.

Dear reviewer, thank you for the comments .There is no stat significance, the percentage of hyperinflators is similar in both groups. 

• Authors conclude patients without stress LVDD had a better exercise tolerance, however this was only assessed in univariate assessment, it would be interesting to correct for age and COPD severity.

Dear reviewer, thank you for the comments.The aim of the study is markers for stress LVDD, if you think it is very important we can give this info in a separate table and section for discussion.

• If COPD severity is different between the groups (after providing the corrrect data, but based on the current data I think it might be), I would recommend including this in the multivariate analyses.

Dear reviewer, thank you for the comments. As you can see from the table the distribution of grade I and II is even in both groups. 

Discussion

• Minor: high frequency of LVDD = high frequency of stress LVDD

Dear reviewer, thank you for the comments.You are right we have corrected this in the discussion section.

Additional comments

• An external validation to assess predictors of stress LVDD would strengthen the paper. I am especially sceptic about RV diameter as an independent predictor (as this is only when measured parasternal, but not the basal diameter)

Dear reviewer, thank you for the comments.You are right the results should be replicated in a separate cohort in order to be validated. We have commented on this in the limitation section, that is added to the manuscript as an additional section.

---

## [Decision Letter · Decision Letter 1]

11 Nov 2020

PONE-D-20-21967R1

Left ventricular diastolic dysfunction in non-severe chronic obstructive pulmonary disease  – a step forward in cardiovascular comorbidome

PLOS ONE

Dear Dr. Cherneva,

Thank you for submitting your manuscript to PLOS ONE. After careful consideration, we feel that it has merit but does not fully meet PLOS ONE’s publication criteria as it currently stands. Therefore, we invite you to submit a revised version of the manuscript that addresses the points raised during the review process.

The reviewers agree that your manuscript has improved. However, there are some remaining issues that I would like you to address.

We look forward to receiving your revised manuscript.

Kind regards,

Hans-Peter Brunner-La Rocca, M.D.

Academic Editor

PLOS ONE

Reviewers' comments:

Reviewer's Responses to Questions

**Comments to the Author**

1. If the authors have adequately addressed your comments raised in a previous round of review and you feel that this manuscript is now acceptable for publication, you may indicate that here to bypass the “Comments to the Author” section, enter your conflict of interest statement in the “Confidential to Editor” section, and submit your "Accept" recommendation.

Reviewer #1: (No Response)

2. Is the manuscript technically sound, and do the data support the conclusions?

Reviewer #1: Yes

3. Has the statistical analysis been performed appropriately and rigorously? 

Reviewer #1: I Don't Know

4. Have the authors made all data underlying the findings in their manuscript fully available?

Reviewer #1: No

5. Is the manuscript presented in an intelligible fashion and written in standard English?

Reviewer #1: Yes

6. Review Comments to the Author

Reviewer #1: This is the revised version of a paper which had been reviewed previously by the Journal. The authors have performed extensive revisions. Overall, I think that the authors have made a big effort and have appropriately addressed most of the reviewers’ comments. There are still a few issues.

1. It is stated in the methods that patients with pulmonary hypertension defined as peak TRV >2.8 m/s were excluded. At the same time cut-off this listed as a criterion for LV diastolic dysfunction.

2. I had asked to report work rate indexed to body weight. This has not been done.

3. Table 4 is hard to understand. What exactly is the dependent variable? This seems to be a mix of linear and logistic regression. In this type of Table, units must not be reported.

7. PLOS authors have the option to publish the peer review history of their article (what does this mean?). If published, this will include your full peer review and any attached files.

Reviewer #1: No

---

## [Author Response · Author response to Decision Letter 1]

12 Feb 2021

Dear Reviewers,

We apologise for the delay of our revision. Due to the Covid-19 pandemia we were very busy in the clinic and we have missed the deadline for revision. As you can see it is only minor revision. So we beg for your benevolence.

Reviewer #1: This is the revised version of a paper which had been reviewed previously by the Journal. The authors have performed extensive revisions. Overall, I think that the authors have made a big effort and have appropriately addressed most of the reviewers’ comments. There are still a few issues.

1. It is stated in the methods that patients with pulmonary hypertension defined as peak TRV >2.8 m/s were excluded. At the same time cut-off this listed as a criterion for LV diastolic dysfunction.

Dear Reviewer, thank you for your remark. One of the exclusion criteria is pulmonary hypertension, defined as peak TRV >2.8 m/s at rest. So none of our patients had TRV>2,8 m/s at rest. However, during exercise 67/104 patients (64%) had stress induced LVDD. 

According to the guidelines it is considered positive if all of the following three conditions are met during exercise: average E/e’ > 14 or septal E/e’ ratio > 15, peak TR velocity > 2.8 m/sec and septal e’ velocity < 7 cm/sec.”Recommendations for the Evaluation of Left Ventricular Diastolic Function by Echocardiography: An Update from the American Society of Echocardiography and the European Association of Cardiovascular Imaging. J Am Soc Echocardiogr 2016;29:277-314’’. So it is stress induced TR>2,8 m/s that is obligatory to define stress induced LVDD. 

2. I had asked to report work rate indexed to body weight. This has not been done.

Dear Reviewer, thank you for your remark.We have taken this into consideration this time. After recalculating the results per body weight it became obvious that there is no significant difference in work load achieved by the two groups. This is mentioned in the text and corrected in both text and table.

3. Table 4 is hard to understand. What exactly is the dependent variable? This seems to be a mix of linear and logistic regression. In this type of Table, units must not be reported.

Dear Reviewer, thank you for your remark.We have subdivided the table into table .4. (presenting the correlation analysis) and table .5. presenting the multivariate logistic regression analysis.We have deleted the units as you recommend.

Table.4. Correlation analysis between stress LV E/e’ ratio and the biomarkers, cardio-pulmonary parameters cut-off values of the echocardiographic idices 

Correlation analysis p-value correlation coefficient

Echocardiographic parameters 

LV E/A ratio rest 0.023 0.616

RV parasternal diameter 0.000 0.793

RVWT 0.000 0.219

RAVI 0.000 0.875

RV E/A ratio rest 0.000 0.417

CPET parameters 

Peak Load 0.730 0.957

Peak VE 0.287 0.613

V'O2 0.048 0.574

AT, V'O2 0.021 0.216

RER 0.943 0.452

VE/VCO2 slope 0.026 0.612

HR at rest 0.737 0.247

Peak HR 0.382 0.409

CRI 0.061 0.752

O2 pulse 0.032 0.481

HRR at 1 min 0.041 0.763

BR, % 0.983 0.213

ICdyn 0.037 0.043

Biomarkers 

PG E2 0.041 0.038

Table.5. Multivariate regression analysis between stress LV E/e’ ratio and the cut-off values of the echocardiographic idices 

Multivariable logistic regression analysis p-value OR 95% CI

RV parasternal diameter 0.001 19.567 3.131-22.290

RAVI 0.000 24.061 4.485-29.100

RV E/A ratio 0.007 10.853 1.913-21.564

---

## [Editor Report · Decision Letter 2]

17 Feb 2021

Left ventricular diastolic dysfunction in non-severe chronic obstructive pulmonary disease  – a step forward in cardiovascular comorbidome

PONE-D-20-21967R2

Dear Dr. Cherneva,

We’re pleased to inform you that your manuscript has been judged scientifically suitable for publication and will be formally accepted for publication once it meets all outstanding technical requirements.

Kind regards,

Hans-Peter Brunner-La Rocca, M.D.

Academic Editor

PLOS ONE
---

## [Editor Report · Acceptance letter]

26 Feb 2021

PONE-D-20-21967R2 

Left ventricular diastolic dysfunction in non-severe chronic obstructive pulmonary disease – a step forward in cardiovascular comorbidome 

Dear Dr. Cherneva:

I'm pleased to inform you that your manuscript has been deemed suitable for publication in PLOS ONE. Congratulations! Your manuscript is now with our production department. 

Kind regards, 

on behalf of

Dr. Hans-Peter Brunner-La Rocca 

Academic Editor

PLOS ONE